# Nutritional Interventions in Pancreatic Cancer: A Systematic Review

**DOI:** 10.3390/cancers14092212

**Published:** 2022-04-28

**Authors:** Aline Emanuel, Julia Krampitz, Friederike Rosenberger, Sabine Kind, Ingeborg Rötzer

**Affiliations:** 1Division of Nutrition Sciences, German University of Applied Sciences for Prevention and Health Management (DHfPG), 66123 Saarbruecken, Germany; a-emanuel@dhfpg-bsa.de; 2Division of Psychology and Pedagogy, German University of Applied Sciences for Prevention and Health Management (DHfPG), 66123 Saarbruecken, Germany; j-krampitz@dhfpg-bsa.de; 3Institute of Psychology, University of Innsbruck, 6020 Innsbruck, Austria; 4Department of Medical Oncology, National Center for Tumor Diseases (NCT), Heidelberg University Hospital, 69120 Heidelberg, Germany; friederike.rosenberger@nct-heidelberg.de; 5Division of Health Sciences, German University of Applied Sciences for Prevention and Health Management (DHfPG), 66123 Saarbruecken, Germany; s-kind@dhfpg-bsa.de; 6Clinic for Oncology and Haemotology, Northwest Hospital, UCT-Cancer University Center, 60488 Frankfurt am Main, Germany

**Keywords:** nutritional interventions, pancreatic cancer, cachexia, malnutrition, weight loss

## Abstract

**Simple Summary:**

This systematic review investigates the impact of nutritional interventions on cachexia, malnutrition and weight loss in patients with pancreatic cancer. In total, 26 studies were included. Parenteral nutrition is associated with a higher incidence of complications. Enteral nutrition shows positive effects on length of stay in hospital, complications, weight loss and cytokines. Dietary supplements enriched with omega-3 fatty acids improve body weight and lean body mass. Considering the heterogeneous study situation as well as the high bias potential of the included RCTs, a recommendation for enteral nutrition and dietary supplements with omega-3 fatty acids can be given.

**Abstract:**

(1) Background: Pancreatic cancer (PaCa) is directly related to malnutrition, cachexia and weight loss. Nutritional interventions (NI) are used in addition to standard therapy. The aim of this systematic review is to provide an overview of the types of NI and their effects. (2) Methods: We included RCTs with at least one intervention group receiving an NI and compared them with a control group with no NI, placebo or alternative treatment on cachexia, malnutrition or weight loss in patients with PaCa. Any available literature until 12 August 2021 was searched in the Pubmed and Cochrane databases. RCTs were sorted according to NI (parenteral nutrition, enteral nutrition, dietary supplements and mixed or special forms). (3) Results: Finally, 26 studies with a total of 2720 patients were included. The potential for bias was mostly moderate to high. Parenteral nutrition is associated with a higher incidence of complications. Enteral nutrition is associated with shorter length of stay in hospital, lower rate and development of complications, positive effects on cytokine rates and lower weight loss. Dietary supplements enriched with omega-3 fatty acids lead to higher body weight and lean body mass. (4) Conclusions: Enteral nutrition and dietary supplements with omega-3 fatty acids should be preferred in nutritional therapy of PaCa patients.

## 1. Introduction

Pancreatic cancer (PaCa) has a poor overall prognosis, with a 5-year survival rate of 9% and a median survival time of 6 months [1]. However, patients with early tumour resection have a higher chance of remission, with a 5-year survival rate of 25% and a median survival of 14–20 months [2,3,4]. Patients often suffer from tumour cachexia [5,6], which is manifested in weight loss [7,8], malnutrition [9,10,11,12] and systemic inflammation [13]. A large proportion of patients have already lost 10% of body weight at initial diagnosis [7,8]. There is an association between malnutrition and, among other things, poorer quality of life [14], postoperative complications [15,16] and higher mortality risk [15]. A prospective multicentre cohort study showed that 71% of PaCa patients had cachexia at time of diagnosis, but only 56% of them received nutritional counselling [17]. Therefore, it is an important goal to improve the nutritional status and so prevent or counteract cachexia [18].

Cachexia is not consistently defined in the literature. It is often defined as weight loss ≥5% in ≤12 months (or body mass index (BMI) <20 kg/m²) and the presence of three of the following five criteria: decreased muscle strength, fatigue, anorexia, low fat-free mass index, abnormal biochemistry (increased inflammatory markers (CRP, IL-6), anaemia (Hb < 12 g/dL), low serum albumin (<3.2 g/dL)) [19]. Independent of cachexia, a weight loss ≥5% caused by malnutrition is also seen to be significant [20,21,22,23,24]. Malnutrition is defined as a weight loss >5% within past 6 months or 10% beyond 6 months; mild to moderate loss of muscle mass; reduced food intake ≤50% of estimated requirements for >1 week or any reduction of >2 weeks; or the presence of chronic gastrointestinal disease that negatively affects food intake or absorption [25]. Furthermore, a chronic disease such as cancer or other diseases must be present [25]. However, the crucial criterion to distinguish cachexia from disease-related malnutrition is the inflammatory process [26]. The systemic inflammation is triggered by cytokines and tumour-derived factors. Increased serum CRP, IL6 and growth differentiation factor-15 are associated with cachexia [19,27]. Appendix A shows factors of tumour cachexia. Malnutrition in the preoperative setting of gastrointestinal cancers has been identified as an independent risk factor for mortality [28]. Weight loss can lead to immunodeficiency, which in turn increases the risk of infection [29].

Different nutritional interventions (NI) exist to counteract cachexia or malnutrition in patients with PaCa. These are parenteral nutrition (PN), enteral nutrition (EN), dietary supplements (DS) and mixed or special forms. In PN, nutrients are applied directly into the bloodstream, bypassing the digestive tract, thereby supplying them to the metabolism [30] (p. 555). In EN, a special liquid food mixture is given through a tube into the gastrointestinal tract. DS are pills, capsules, tablets, powder, sip feed nutrition or liquids to supplement diet. The individual components are part of the normal diet and are intended to supplement it if they are not sufficiently available for certain reasons [30] (p. 128). Mixed or special forms are NI that are composed of different interventions mentioned above or represent special forms such as nutritional counselling or fasting. These are used before (pre), during (peri), after (post) or even without connection to a surgical procedure.

The current data situation regarding NI in cachectic patients is referred to as “wildfire” [31]. Comprehensive reviews of different types of interventions and their impact on malnutrition and cachexia in PaCa are few and sometimes only address specific nutrients such as omega-3 fatty acids [32,33,34,35]. A narrative review focusing on the efficacy of NI in cachectic PaCa patients concluded that nutritional counselling, to increase energy and protein intake, should be the first step, followed by DS and PN [36]. Two reviews in gastrointestinal cancer/PaCa patients examined the development of postoperative infectious complications and the postoperative and immunology outcome parameters and showed a significant reduction in postoperative infectious complications and shorter length of stay in the hospital with preoperative immunonutrition [37,38]. Another systematic review in patients undergoing pancreaticoduodenectomy focused on the perioperative nutritional supplementation concluded that attention should be paid to preoperative nutritional optimization and nutritional support should be delivered in the postoperative period with cyclical enteral nutrition [32]. A narrative review in patients with cancer-associated cachexia focused on a diet in relation to the treatment of cancer-associated cachexia and shows that a multimodal therapy consisting of a combination of exercise, nutrition and/or other measures seems to be gaining importance [39]. This interdisciplinary approach is confirmed by another review looking at patients with cancer cachexia and focusing on the diagnostic criteria and therapeutic approaches of cancer cachexia [40]. The ESPEN Guidelines recommend nutritional assessment for all cancer patients at diagnosis and provision of appropriate nutritional support (preferably nutritional counselling followed by artificial nutrition (EN, PN)) according to need [41]. Altogether, the body of literature on the effects of different NI in patients with PaCa is complex, heterogeneous and often unclear. 

Therefore, the aim of this systematic review was to summarize the effects of different NI (PN, EN, DS, mixed or special forms) in PaCa patients on cachexia, malnutrition and weight loss. Because of the poor data situation regarding NI in PaCa, parameters associated with the nutritional status such as immunology and complications should be included. Furthermore, recommendations for clinical practice should be derived if appropriate.

## 2. Materials and Methods

This systematic review was prepared following the PRISMA guidelines [42] as well as the Cochrane collaboration guidelines [43]. The protocol was registered in PROSPERO database (CRD42021270502) on 29 August 2021. 

### 2.1. Search Strategy

All available literature (from 1957) to 12 August 2021 was searched in Pubmed and Cochrane library databases by two independent researchers. A hand search of the bibliographies of the selected papers was also conducted. The search strategy was defined using the PICO scheme [44]:P(opulation): pancreatic cancer patients;I(ntervention): nutrition intervention (PN, EN, DS, mixed or special forms);C(omparison): no nutrition intervention or placebo or alternative treatment;O(utcome): malnutrition or cachexia or weight loss.

Malnutrition and cachexia were defined as a weight loss or BMI < 20 kg/m². Furthermore, criteria closely related to cachexia or malnutrition (e.g., decreased muscle strength, fatigue, anorexia, low fat-free mass index, abnormal biochemistry (increased inflammatory markers (CRP, IL-6), anaemia (Hb < 12 g/dL), low serum albumin (<3.2 g/dL) [19]) were also included.

Due to the complex definitions of cachexia, malnutrition and weight loss, surrogate parameters to nutritional status (weight loss and parameters of bioelectrical impedance analysis (BIA)), immunology (cytokines, C-reactive protein (CRP), albumin) and complications (infections, mortality, length of stay in hospital) were used as additional outcomes in order to represent a large database.

The search term was (Nutrition* OR Diet* OR Supplement*) AND ((Pancrea* Cancer) OR (Pancrea* Carcinoma) OR (Pancrea* Tumour) OR (Pancrea* Neoplasm)) AND (Malnutrition OR Cachexia OR (Weight Loss)).

### 2.2. Eligibility Criteria

We included studies in humans with PaCa (incl. periampullary tumours) or several types of cancer including PaCa. The studies needed to be randomized controlled trials (RCTs) with at least one intervention group receiving an NI and compared with a control group with no NI, placebo or alternative treatment on cachexia, malnutrition or weight loss and their surrogate parameters described above. Further inclusion criteria were: patients ≥ 18 years of age, published in full text in English, of all publication years. 

Studies were excluded if pancreatic enzyme replacement therapy was the only NI or just the risk of developing PaCa was investigated. Duplicates were also excluded.

### 2.3. Study Selection

The studies were imported in Citavi literature management program. The two independent reviewers (A.E. and J.K.) screened the titles and abstracts and in the next step the full texts to filter them against inclusion and exclusion criteria.

### 2.4. Quality Criteria

Primary endpoints of included RCTs were checked for quality criteria by two independent reviewers (A.E. and J.K.) using the Cochrane Collaboration tool (RoB2) for assessing bias risk [43]. In case of disagreement, a third reviewer was consulted for assistance (S.K.). No studies were excluded based on quality assessment.

### 2.5. Data Extraction and Analysis

The following data were extracted from the full texts: entity(ies), sample characteristics, description of nutrition intervention(s), outcome(s), results.

Due to considerable statistical and methodological heterogeneity (e.g., different length and type of intervention), a random-effect meta-analysis could not be performed. The studies were sorted by NI (PN, EN, DS, mixed and special forms). The intervention nutritional counselling was integrated in mixed and special forms, as the current study situation shows a trend towards individualized nutritional counselling.

Outcomes are presented as reported by the authors.

## 3. Results

### 3.1. Overview

A Prisma flow diagram is given in Figure 1. In Pubmed, 43 trials were found, and in the Cochrane database, 144 trials were found. After a duplicate check, 31 were excluded. Titles of 156 papers were screened, of which 129 were excluded due to inappropriate subject matter. This left 27 trials from which the abstracts were checked. Ten abstracts were excluded because they were thematically unsuitable. Seventeen full texts were reviewed, and two trials were excluded because of the randomization scheme and no difference between NI in intervention and control group. Fifteen trials were included from the database search.

Further, 11 trials were included through a manual search of the bibliographies of the searched trials, so that a total of 26 RCTs were considered in the review.

All trials are presented in tabular form. They are described in more detailed in the text and assessed according to risk of bias as described above.

The characteristics of the included studies are given in Table 1. The systematic review included 2720 patients across the selected studies (PN: *n* = 277; EN: *n* = 981; DS: *n* = 528; mixed/special forms: *n* = 934). The average age was 65 years (PN: 62 years; EN: 64 years; DS: 68 years; mixed/special forms: 68 years). Of all of the subjects, 57% were male (PN: 56%; EN: 64%; DS: 53%; mixed/special forms: 51%). The average BMI was 22.5 kg/m² (reported in 54% of the included studies) (PN: 26.0 kg/m²; EN: 21.3 kg/m²; DS: 22.3 kg/m²; mixed/special forms: 23.3 kg/m²). In terms of entities, nine studies (35%) looked at PaCa alone. Fourteen studies (54%) were included that looked at other cancer types in addition to PaCa. Three studies (11%) looked at periampullary tumours. Of mixed-entity studies, all results are reported.

### 3.2. Parenteral Nutrition

Three RCTs were included that did not have common outcomes (see Table 2) [51,60,63]. The timing of intervention ranged from 2 days before to 5 days after surgery. 

Regarding effects on our category nutritional status, one RCT investigated the effects of PN adapted with vitamins and trace elements on fasting days vs. an isotonic electrolyte solution on fasting days on body weight and weight gain [63]. There was no difference in the weight change from hospital admission to discharge between groups when medians were compared (*p* = 0.217). However, a multivariate regression analysis adjusted for nine potential influencing factors revealed a significant advantage for the intervention group (*p* = 0.027).

Two studies examine the impact of parenteral nutrition on the occurrence of complications according to our definition from Brennan et al. (1994), who compared total parenteral nutrition (TPN) with macro- and micronutrients in an intervention group with dextrose-containing saline (without TPN) to a control group on survival, complications, mortality, reoperation and length of stay in hospital. There was no significant difference in median survival between groups (*p* = 0.25). Further, there was no significant difference in minor complications between groups (*p* = 0.30). However, major complications and severe complications occurred more often in the intervention group compared to the control group (*p* = 0.02 and *p* = 0.01). PostOP mortality (*p* = 0.17) and reoperation rate (*p* = 0.18) were not significantly different between groups. The authors concluded that there is no justification for administration of a TPN after pancreatic resection. 

The second trial in our category complications investigated the effects of a parenteral amino acid supplement (+glutamine) vs. an isonitrogen parenteral amino acid supplement on patients after pancreaticoduodenectomy [60]. They concluded that median postOP length of stay in hospital was similar between groups (*p* = 0.197). The study was stopped halfway through because the glutamine solution was no longer available. 

In summary, parenteral nutrition appears unlikely to lead to weight change and seems to result in a higher chance of postoperative complications. 

### 3.3. Enteral Nutrition 

Ten RCTs with EN could be included (see Table 3) [49,52,55,56,58,61,62,64,65,69]. Studies were conducted very differently with regard to the start of the intervention which ranged from 14 days before to 16 weeks after surgery.

In our category nutritional status, one trial calculated EN according to Harris and Benedict’s energy requirements vs. nutritional counselling by a dietician with recommendations on energy and protein requirements. The intervention group showed a lower weight loss vs. control group after 2 months (*p* =0.0031) [56].

In our category immunology, two trials looked at cytokines [58,69]. The first trial investigated postoperative enteral immunonutrition with glutamine and arginine vs. standard EN [69]. They showed higher concentrations of IL-6 (on day 10 postOP *p* = 0.017), IL-8 (on day 1 postOP: *p* = 0.01; on day 3, 7, 10 postOP: *p* < 0.001), IL-10 (on day 3 and 10 postOP: *p* < 0.001), IL-1RA (on day 7 postOP: *p* < 0.001; on day 10 postOP: *p* = 0.002) and a significant increase in total lymphocyte count (*p* = 0.003) in the intervention group compared to the control group. The control group showed significantly higher levels of proinflammatory cytokines such as IL-1ß (on day 7 postOP: *p* < 0.001) and TNF-α (on day 3 postOP: *p* = 0.006; on day 7 postOP: *p* < 0.001). The increase in total lymphocyte count indicates a better nutritional status and less malnutrition in the intervention group. The second trial [58] showed, after 14 days of preoperative standard nutritional supplementation with Fresubin^®^ vs. immune-boosting nutrition Impact^®^, no significant difference in IL-1-α (*p* = 0.529) and TNF-α (*p* = 0.112) between groups, but a significant decrease in TNF- α in the intervention group over period of 14 days (day 14 preOP until day of OP) (*p*= 0.047).

In our category complications, seven RCTs looked at postoperative complications (+length of stay in hospital [49,52,55]). Braga (1999) and Daly et al. (1995) compared an enteral formula diet as a standard diet with an enteral formula and a complementary diet. In Braga (1999), this diet was with a supplement of arginine, RNA as well as omega-3 fatty acids. In Daly et al. (1995), this comparison was conducted during hospitalization and afterwards in the outpatient sector compared with no outpatient therapy. Braga (1999) showed that postOP infectious complications occurred more frequently (*p* = 0.02) and length of stay in hospital was longer (*p* = 0.01) in the control group compared to the intervention group. Daly et al. (1995) showed significant frequent development of major postoperative infections/ wound complications (*p* < 0.005) and longer length of stay in hospital (*p* <0.05) in the control group. Gade et al. (2016) compared preoperative immunonutrition with no nutritional therapy (clinical standard). There was no significant difference in postoperative complications (*p* > 0.05). The authors showed a significantly larger number of participants with more than three postoperative complications in the control group (*p* = 0.030). For the second primary outcome, length of stay in hospital, there was no significant difference between groups (*p* = 0.549). A further trial with fat-free elemental feeding via a tube (**IG**: 3 months postOP vs. **CG**: until oral intake was achieved) showed that the cumulative readmission rate at 3, 6, 12 and 24 months postOP was significantly lower in the intervention vs. control group (*p* = 0.018), which did not receive long-term nutrition therapy [65]. There was a significant increase in readmission rate in the control group vs. intervention group at 90 days (*p* = 0.047), 6 (*p* = 0.044) and 12 months (*p* = 0.065) postOP. Further trials compared an immunostimulatory EN after surgery with a standardized EN [61,64] (+ isonitrogenous, isocaloric diet [64]) on postoperative complications/infectious complications. Klek et al. (2008) showed no significant differences in postoperative complications between groups (*p* > 0.05). The authors stated that despite adequate patient compliance, immunostimulatory EN had no effect on the primary outcome. Lobo et al. (2006) reported no significant differences in infectious complications (*p* = 1.0). The authors conclude that there is no outcome benefit for patients. Klek et al. (2011) [62] showed a significantly lower rate of infectious complications in the intervention group with enteral immunonutrition compared to standard enteral nutrition (*p* = 0.04). 

In summary, nutritional status and especially weight loss was significant lower after 2 months of EN vs. a nutritional counselling alone. Cytokine levels significantly differed only in some studies. Infectious complications were lower in some trials when enteral immunonutrition enriched with immunomodulating substances such as omega-3 fatty acids and special amino acids were compared to standardized EN. Some trials could not confirm this and showed no significant differences between groups. Some trials showed significantly lower length of stay in hospital in the intervention group, but not all.

### 3.4. Dietary Supplements 

Seven RCTs in the field of dietary supplements could be included (see Table 4) [46,47,50,53,54,66,70]. The timing of interventions varied from 7 days before until 8 weeks after surgery.

In our category nutritional status, five trials were included. They examined the effects of a dietary fortification with omega-3 fatty acids [46,47,54,66,70]. Three RCTs looked at the effects of a dietary supplement with protein and energy and enriched with omega-3 fatty acids compared to a isocaloric isonitrogenous dietary supplement on body weight or change in body weight [54,66,70]. One trial [54] observed that body weight and lean body mass remained stable and did not differ significantly between groups (*p* = 0.74, *p* = 0.88). Furthermore, this study showed a significant correlation between supplement intake and weight gain (*p* < 0.001) and increase in lean body mass (*p* = 0.036) within the intervention group. Another trial looked at body weight and body composition with a dietary supplement high in protein and energy and enriched with omega-3 fatty acids vs. an isocaloric isonitrogenous supplement without omega-3 fatty acids [47]. There was no significant difference in weight change during the eight weeks of intervention (*p* = 0.052). Another trial looked at weight change [70] compared to omega-3 fatty acids from fish oil or from marine phospholipids and showed no difference between the groups. Nevertheless, a significant weight stabilization after 6 weeks in comparison to the weight loss before the study within both groups was observed (IG: *p* = 0.001, CG: *p* = 0.003). The third trial [66], which compared dietary supplements with and without omega-3 fatty acids, was also unable to show any significant differences in change in body weight and lean body mass between groups (p reported as NS). One older trial compared a high- with a low-calorie standard diet (IG: 2000 kcal protein enriched diet vs. CG: 1300 kcal) on weight loss/weight change [53]. They reported that weight loss was lower in the intervention group (*p* not reported) when only pancreatic patients were considered but weight change during therapy was higher in the intervention group when all patients were considered, because weight loss in patients with gastric and colorectum cancer were higher (*p* not reported). 

In our category immunology, two trials were included [46,50]. The first looked at serum concentration of IL-6 with a preOP EPA fortified diet (dietary oral supplement) with 1200 kcal vs. a standard diet, isocaloric, isonitrogenous without EPA, and showed no significant difference (*p* = 0.68) between groups after 7 days of intake. The second trial [50] observed the effects of a preoperative oral supplement containing glutamine, antioxidants and green tea extract vs. a placebo (based on orange juice) on inflammatory response. C-reactive protein levels (postOP) did not change significantly within groups (*p* = 0.52; *p* = 0.13; *p* = 0.57; *p* = 0.97) [50].

Altogether, none of the reported trials showed effects of dietary supplements on weight change. One trial showed a correlation between supplement intake and weight gain and increase in lean body mass. Further trials showed no significant differences in lean body mass. A preOP diet (enriched with EPA or glutamine) showed no significant differences on IL-6 and C-reactive protein levels.

### 3.5. Mixed and Special Forms of Nutritional Interventions

In the category of mixed and special forms, six RCTs could be included (see Table 5) [45,48,57,59,67,68]. The timing of the intervention was dependent on the type of nutritional intervention and ranged from 5 days before surgery to 3–4 months of general care during chemotherapy.

In our category nutritional status, all trials looked at weight change, ratio of skeletal muscle mass and the recovery of nutritional state. The first trial [67] compared EN with total PN on weight change and showed significantly higher BMI on postoperative day 21 (*p* = 0.005) in the group with enteral nutrition compared with total PN. However, the BMI preOP was slightly higher in the EN group compared to total PN (EN: 23.8 kg/m²; TPN: 23.5 kg/m²). The second trial investigated a combination of an enteral supplement (enriched with EPA) and three nutritional consultations during radiotherapy vs. a normal diet [45]. They showed no significant difference in the ratio of skeletal muscle mass and psoas major muscle area (*p* = 0.102) after intervention during neoadjuvant chemotherapy. A further trial with two intervention groups and one control group compared a standard EN (+electrolyte solution) (IG1) and a parenteral nutrition with vitamins and trace elements (+electrolyte solution) (IG2) with an oral supportive nutrition (+electrolyte solution) (CG) on the recovery of nutritional status. They showed no significant differences in body weight (*p* reported as NS), body fat (*p* reported as NS) and lean body mass (*p* reported as NS) over the time of 12 months between groups [59]. 

In our category immunology, one trial was included [57] which looked at the difference between preoperative enteral immunonutrition (IG1), enteral immunonutrition plus glycine (IG2) and no preoperative nutritional support in a control group over different time periods on postoperative serum level of C-reactive protein. The authors showed significantly lower levels in both the intervention and control group on postoperative day 7 (*p* < 0.05). However, there was a significant increase in C-reactive protein levels in all groups vs. baseline (*p* > 0.05).

In our category complications, two trials were included. The first trial looked at the effects of nutritional counselling vs. usual supply on 1-year mortality [48]. They concluded that there was no significant difference in 1-year mortality between groups (*p* = 0.74). The second trial compared PN with EN and showed a significant lower incidence of postoperative complications (*p* = 0.040) but no significant difference in incidence of infectious complications (*p* = 0.731) in the intervention vs. control group [68].

In summary, there were beneficial effects of EN compared to PN on body mass index on day 21 postOP but no beneficial effects of EN or PN compared to the electrolyte solution on body weight, body fat and lean body mass when the period of 12 months was considered. A preoperative EN with a supplemental immunonutrition with glycine belongs to significant lower serum levels of C-reactive protein on POD7 vs. no supplement. One trial show significantly lower incidence of postoperative complications in EN vs. PN.

### 3.6. Bias

As figured in the overall overview of the RCT studies (Figure 2), the majority of the included RCT studies have a high or moderate risk of bias. Eleven studies were assigned a high risk of bias in the overall assessment. Eleven studies have some concerns/problems and thus an unclear risk (some concerns). Four studies indicate a low risk of bias. Especially in the categories “randomization process” and “missing outcome data”, there was more frequent assessment with a high risk of bias.

## 4. Discussion

The aim of this systematic review was to describe the effect of a nutritional intervention compared to no nutritional intervention, a placebo or an alternative treatment in PaCa patients on cachexia, malnutrition and the resulting weight loss. The primary outcomes of the included 26 RCTs were highly heterogeneous. The most common primary outcomes were complications and body weight/weight loss.

Some of the included studies were hard to interpret. For example, one RCT in the category PN showed no effect on median weight change but a positive effect in weight change adjusted for nine potential influencing factors in a multivariate regression analysis. In a multivariate analysis, nutritional intervention increased body weight particularly in patients with malignant lesions [63]. This means that the groups were highly heterogeneous, which reduces the meaningfulness of the trial. Another trial looked at the generic role of total PN and defined this outcome as a summary of survival, complications, mortality (postOP) and reoperation. This complex definition made the interpretation of the study difficult. They showed no effects on survival, minor complications, mortality and reoperation but negative effects on major and severe complications in the intervention group. The authors concluded that routine administration of PN after surgery in patients undergoing pancreatic resection is not justified [51]. These results are in line with another trial, which showed no effect on length of stay in hospital [60]. This is also confirmed by the bias assessment, which rated two studies in the category of PN with “some concerns” and one study with a high risk of bias. Based on the present data, we would not recommend routine use of PN because of the frequent complications.

RCTs from the category EN showed that EN and complementary/enriched enteral immunonutrition must be differentiated. Most of the included RCTs in the EN group examined the effects of supplemental immunonutrition [49,52,55,58,61,62,64,69]. Supplements often include arginine, RNA and omega-3 fatty acids. Three studies showed positive effects of supplemental immunonutrition on postoperative complications [49,52,62] compared to the control group, while further, three trials found no difference [55,61,64]. Two trials showed a positive effect on length of stay in hospital [49,52], while one trial showed no effect [55]. One trial showed a positive effect on proinflammatory cytokines [69], and one trial showed no effect between groups but a positive effect on TNF-α in the intervention group over period of 14 days [58]. The change in cytokine and total lymphocyte is associated with an immunomodelling effect and better nutritional status. One trial showed a positive effect on weight loss [56]. In summary, some trials showed positive effects and some trials no effect of EN on our outcome categories nutritional status, immunology and complications. There were no trials with a negative effect of EN on our defined outcomes. In the bias assessment, the studies in the category EN were frequently rated as “some concerns” (five times) and “high risk” (four times). Only one study was rated as low risk. Therefore, EN can only be recommended as a possible adjuvant therapy for pancreatic cancer. Further studies of higher quality are necessary in order to derive effects more clearly and to be able to make recommendations.

There were several studies in the category of dietary supplements that investigated an oral supplement that was enriched with omega-3 fatty acids in the intervention group. There was one trial that looked at weight loss which showed a positive effect [53], and another trial that showed no effect [54]. One trial showed a positive effect on body weight after 8 weeks [47], and one trial which showed no effect [66]. A positive effect on weight gain and lean body mass was shown in correlation to supplement intake [54]. There were three trials which showed no effect on lean body mass [47,54,66]. One trial showed a positive effect of weight stabilization [70]. One trial showed no effect of a dietary supplement on serum concentration of IL-6 [46]. A further trial showed no effect on C-reactive protein levels [50]. The studies show that a heterogeneous picture prevails with regard to dietary supplements. This trend was to be expected in the studies conducted. There was a heterogeneous picture in the bias assessment of studies investigating dietary supplements (two trials with “low”, two trials with “some concerns” and three trials with high risk of bias). The overview of the risk of bias (Figure 2) shows that the studies have different weaknesses. Often there were weaknesses in the randomization process. Further, the results of the trials are dependent on various factors. For example, many studies did not ensure that the intended nutritional target was achieved. The effect of an individual calculation and control of the adherence to the dietary intervention could be investigated in future studies. RCTs comparing different interventions were also included. In three RCTs, EN was compared with PN [59,67,68]. One trial that looked at EN vs. PN showed a positive effect on BMI on day 21 postOP [67]. One trial that looked at a preoperative dietary supplement as immunonutrition (+glycine in second intervention group) compared to no preoperative nutritional support showed a positive effect on serum levels of C-reactive protein at day 7 postOP [57]. A further trial looked at PN compared to EN and showed a positive effect on the incidence of postOP complications but no effect on infectious complications [68]. There were further trials which showed no effect on the reported outcomes. In our defined category nutritional status, there was one trial that looked at enteral immunonutrition combined with nutritional counselling vs. a normal diet and showed no effect on the ratio of skeletal muscle mass and psoas major muscle mass [45]. One trial compared EN in intervention group one and PN in intervention group two with a standard electrolyte solution in a control group and showed no effect on body weight, body fat and lean body mass after 12 months [59]. One further trial in the category complications compared nutrient supply (energy, protein) combined with nutritional counselling with nutrient supply alone, and showed no effect on 1-year mortality [48]. The bias assessment of the studies that investigated mixed interventions showed a trend towards high risk for bias (one trial with “low”, two trials with “some concerns” and three trials with high risk of bias). In summary, mixed interventions were hard to interpret and no recommendation can be derived from this. Further trials should focus on simple or multiple nutritional intervention compared with a control group. An evaluation of nutritional intervention is generally difficult and subject to confounding factors, as compliance is often not fully assessed. With regard to nutritional counselling, compliance is very individual and depends on the counsellor–client situation. In the presented studies, the actual total amount of consumed nutrients/food was often not stated. A dose–response relationship therefore cannot be derived.

A comparison of the results of this review with other reviews from this field shows heterogeneous results. A narrative review focusing on the efficacy of nutritional interventions in cachectic PaCa patients concluded that nutritional counselling to increase energy and protein requirements should be the first step, followed by dietary supplements and PN [36]. This is not in line with our results. We conclude in our review that nutritional counselling should be given as a complementary measure and EN or immunonutrition with dietary supplements should be prioritized. In line with our results are two reviews which showed a positive effect in gastrointestinal cancer/PaCa patients on postoperative infectious complications and shorter length of stay in hospital with preoperative immunonutrition [37,38]. Furthermore, in line with our results is another systematic review in patients undergoing pancreaticoduodenectomy focused on the perioperative nutritional supplementation, which concluded that attention should be paid to preoperative nutritional optimization and nutritional support should be delivered in the postoperative period with cyclical enteral nutrition [32]. In addition to the results of this review, modern supports through in-app interventions have also shown success in terms of nutritional status in an RCT with PaCa patients [71]. A further recent analysis demonstrated a clear advantage in median survival of EN or PN over oral nutrition in advanced cancer-related cachexia [72].

The risk of bias assessment showed a large number of RCTs with high and moderate risk of bias. From 26 studies, 11 were rated as high risk and 11 as medium risk. Only four studies had a low risk of bias according to Cochrane’s RoB2 assessment. The impact of the data is therefore generally difficult. The high number of studies assessed as high risk may be attributable to the high standard of the assessment tool. Due to the impracticability of the meta-analysis and the overall high risk of bias in the quality assessment, the results should be treated with caution. Furthermore, previous studies rarely used placebos in control groups. This can lead to bias because the control group does not expect positive effects and may not receive the same attention by study personnel. Although a large number of studies were included in this review, the sample sizes of the individual studies were very heterogeneous and often very small. Underpowered studies can also lead to bias in results, especially because no meta-analysis could be calculated. Likewise, bias in results can occur because studies with different entities were included if they contained PaCa. A separate analysis was not reported. An analysis of studies that only included PaCa patients would have minimized the risk of bias with regard to the entities but would have greatly minimized the number of studies to be included. Many studies were analysed by intention-to-treat-analysis, which appears appropriate, but it should be noted that the intervention might have larger effects if followed successfully. The timing of the start of the interventions was also heterogeneous. This makes direct comparisons between studies difficult.

There are limitations that should be taken into consideration. The most important limitation lies in the heterogeneous outcomes. Therefore, a meta-analysis could not be calculated. Furthermore, the search in Pubmed did not lead to a complete result. This is shown by the high number of studies that were added to the review by looking at the bibliographies and by hand searching. Searching more databases and using mesh terms could have improved the search. The heterogeneous literature and number of trials with high risk of bias were further limitations.

The greatest strength of our review is that different forms of nutritional interventions were considered. This gives the reader a good overall view regarding the effects of the nutritional interventions. The entire research process of two reviewers followed the Prisma Guidelines and the standards of the Cochrane Handbook in order to conduct systematic reviews throughout the process. In case of discrepancies, a third independent person was consulted.

In addition to certain amino acids and omega-3 fatty acids, in future trials more attention should also be paid to micronutrients via individual blood levels, so that the focus is not only on macronutrients and on calorie balance. This is because micronutrients such as selenium and copper also show a direct influence on the development of PaCa and survival time [73]. In order to have a larger database in the field of nutrition interventions and to be able to derive evidence-based recommendations, more RCTs on PN, EN and dietary supplements should be conducted. Further, RCTs on anti-inflammatory supplements and anti-inflammatory diets should be planned.

## 5. Conclusions

This systematic review investigated the effects of different nutritional interventions to treat cachexia, malnutrition and weight loss in patients with PaCa. PN was associated with frequent complications. EN showed positive effects on length of stay in hospital, complications, cytokines and weight loss. Dietary supplements enriched with omega-3 fatty acids maintained and possibly increased body weight and lean body mass. After weighing the advantages and disadvantages of each intervention, an individualized diet should be administered depending on the patient’s condition or whenever possible consisting of EN or dietary supplements as immunonutrition enriched with omega-3 fatty acids and specific amino acids. However, due to the heterogeneity of the data, clear recommendations for these interventions are not possible.

## Figures and Tables

**Figure 1 cancers-14-02212-f001:**
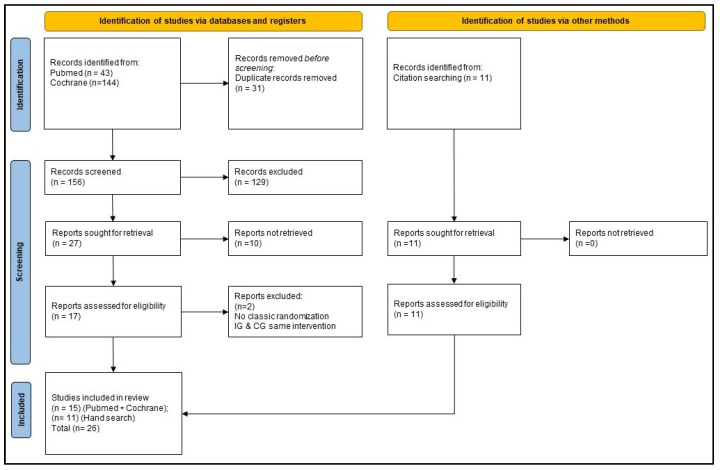
Prisma flow diagram.

**Figure 2 cancers-14-02212-f002:**
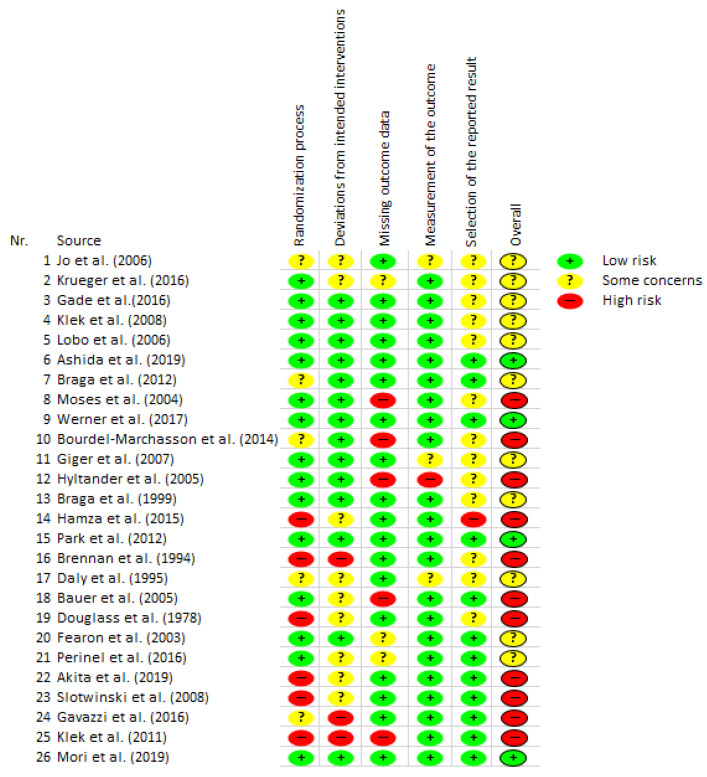
Risk of bias (RCTs via RoB2 by Cochrane).

**Table 1 cancers-14-02212-t001:** Characteristics of included studies.

Reference	Type of Cancer and Sample Size	Patients’ Characteristics	Intervention	Intervention Time Point/ Duration	Primary Outcome(s)
Akita et al. (2019) [45]	PaCa *n* = 62	**IG** (*n* = 31): ♂11; ♀20 67.8 (±10.7) years 22.3 (±2.39) kg/m² **CG** (*n* = 31): ♂16; ♀15 66.4 (±9.8) years 22.0 (±3.06) kg/m²	**IG**: normal diet + EPA-enriched EN as food supplement and 3 nutritional consultations (before, during and after radiation). Composition (target): 2 bottles (440 mL): 560 kcal + EPA (Prosure^®^ (Abbott, Japan))/d. **CG**: Normal diet.	During neoadjuvant chemoradiotherapy (approx. 5 weeks)	Ratio of skeletal muscle mass
Ashida et al. (2019) [46]	Periampullary cancer *n* = 20	**IG**: (*n* = 11): ♂5; ♀6 64 (±11) years 55.9 (±13.5) kg 22.3 (±7.6) kg/m² **CG**: (*n* = 9): ♂6; ♀3 69 (±6) years 56.3 (±7.2) kg 21.4 (±2.5) kg/m²	**IG**: dietary supplement (target: 600 kcal/d) with EPA fortified diet (2.0 g/d) + regular diet (1200 kcal). **CG**: standard diet: isocaloric, isonitrogenous standard diet (target: 600 kcal/d) without EPA + regular diet (1200 kcal).	7 days preOP	Serum concentration of IL-6
Bauer et al. (2005) [47]	PaCa *n* = 185	**IG**: (*n* = 87): ♂n.a.; ♀n.a. 66.8 (±1.0) years 62.9 (±1.2) kg 22.4 (±0.4) kg/m² **CG**: (*n* = 98): ♂n.a.; ♀n.a. 68.3 (±1.1) years 59.3 (±1.3) kg 21.2 (±0.4) kg/m²	**IG**: target 2 doses of a dietary supplement high in protein and energy + omega-3 fatty acids (1.2 g EPA). **CG**: isocaloric, isonitrogenous control supplement without omega-3 fatty acids. Both formulas: 310 kcal, 16 g protein. Daily intake Ø 1.5 doses of oral suppl./d (--> 465 kcal and 24 g protein).	Unresectable PaCa; 4–8 weeks	Body composition (body weight, lean body mass)
Bourdel-Marchasson et al. (2014) [48]	Mixed *n* = 336 thereof PaCa *n* = 62	**IG**: (*n* = 169): ♂81; ♀88 77.7 (±5.2) years WL: 8.9 (±6.6)% **CG**: (*n* = 167): ♂91; ♀76 78.3 (±4.7) years WL: 8.6 (±7.9)%	**IG**: usual care + NI: usual nutrient supply + nutritional counselling. Energy target: 30 kcal/kg body weight/d. Protein target: 1.2 g/kg body weight/d. Possibly (if necessary) dietary supplement. **CG**: usual care group: normal nutrient supply everything allowed.	According to duration of chemotherapy; 3–4 months	1-year mortality
Braga et al. (1999) [49]	CoCa, GaCa, PaCa *n* = 171 thereof PaCa *n* = 22	**IG**: (*n* = 85): ♂50; ♀35 60.9 (±11.9) years 65.8 (±10.9) kg **CG**: (*n* = 86): ♂56; ♀30 60.8 (±9.7) years 67.6 (±11.2) kg	**IG**: EN (Impact, Novartis) (1 L/d) (target): 12.5 g arginine, 1.2 g RNA, 3.3 g omega-3 fatty acids. **CG**: similar EN without enrichments Both: isocaloric and isonitrogenic.	6 h postOP–7 days postOP	Rate of postoperative infectious complications and LOS
Braga et al. (2012) [50]	PaCa, periampullary cancer *n* = 36 thereof PaCa *n* not reported	**IG**: (*n* = 18): ♂11; ♀7 64.1 (±10.8) years 25.9 (±4.4) kg/m² WL: 4.4% **CG**: (*n* = 18): ♂12; ♀6 64.1 (±12.6) years 24.2 (±3.8) kg/m² WL: 4.3%	**IG**: dietary supplement as pre-conditioned oral carbohydrate supplement (pONS) enriched with glutamine, antioxidants and green tea extract. Target: 3 doses (first 1 day before surgery at 3 pm, second 6 h later, third on the day of surgery 3 h before induction of anaesthesia); pONS was administered shortly before surgery to have glutamine and antioxidants ready for surgery. **CG**: Placebo drink.	1 day preOP–3 h preOP	Postoperative host’s antioxidant capacity (TEAC) and inflammatory response (CRP)
Brennan et al. (1994) [51]	PaCa, periampullary cancer *n* = 117 thereof PaCa *n* not reported	**IG**: (*n* = 60): ♂34; ♀26 65 (34–86) years WL: 5.8 (0–18)% **CG** (*n* = 57): ♂27; ♀29 63 (30–86) years WL: 6.8 (0–22)%	**IG**: total PN 1 day postOP until day with oral intake >1000 kcal/d (12.3 (6–34) d). Total PN (target): 1 g/kg BW/d protein and 30–35 kcal/kg/d + electrolytes, vitamins, minerals (non-protein energy from 70% glucose, 30% fat). **CG**: dextrose-containing saline until postoperative intake exceeds 1000 kcal/d (22.2 (3–69) d).	1 day postOP until oral intake >1000 kcal/d	Generic role of total PN (postOP mortality and morbidity)
Daly et al. (1995) [52]	OeCa, GaCa, PaCa, others *n* = 60 thereof PaCa *n* = 15	**IG****1 and 2**: (*n* = 30): ♂25; ♀5 61 (±12) years WL: *n* = 12 **CG** **1 and 2**:(*n* = 30): ♂16; ♀14 61 (±10) years WL: *n* = 10	**IG****1** (*n* = 18): enteral IN in hospital and ambulant. **IG** **2**: (*n* = 12): enteral IN only in hospital. **CG** **1**: (*n* = 19): EN with standard diet in hospital and ambulant. **CG** **2**: (*n* = 11): enteral standard diet only in hospital. Patients did not receive oral nutrition for the first 7 days postOP.	1 day postOP –12–16 weeks after diagnosis	Clinical outcome, white blood cell fatty acid composition and PGE_2_ secretion
Douglass et al. (1978) [53]	PaCa, GaCa, CoCa *n* = 30 thereof PaCa *n* = 15	**No characteristics:** *n* = 13 PaCa; *n* = 2 Ampullary or Duodenal Ca; *n* = 5 GaCa *n* = 5 Rectosigmoidal Ca; *n* = 4 RectalCa; *n* = 1 AnalCa	**IG**: standard diet + dietary supplement (3 times/d) (300 mL of chilled flavoured solution (1 kcal/mL)/ d: 900 kcal. **CG**: standard diet.	Before planned radiotherapy; between meals three times/d	Weight loss and weight changes
Fearon et al. (2003) [54]	PaCa *n* = 200	**IG**: (*n* = 95): ♂54; ♀41 67 (±1) years 60.3 (±1.1) kg 21.8 (±0.4) kg/m² WL: 17.9 (±0.9)% **CG**: (*n* = 105): ♂56; ♀49 68 (±1) years 61.4 (±1.2) kg 22.0 (±0.4) kg/m² WL: 17.1 (±0.8)%	**IG**: target of 2 doses of dietary supplement with 310 kcal each, 16 g protein, 6 g fat with 1.1 g EPA and antioxidants. **CG**: target of 2 doses of dietary supplement with 310 kcal each, 16 g protein, 6 g fat and antioxidants.	Advanced PaCa (unresectable) with weight loss; 8 weeks	Body weight and body composition
Gade et al. (2016) [55]	PaCa *n* = 35	**IG**: (*n* = 19): ♂7; ♀12 68 (50–81) years 70.5 (50.8–103.4) kg 24.3 (18.8–28.3) kg/m² WL total: −5.5% (−16.5–2.1) WL last Month: 1.9% (−9.4–2.1) **CG**: (*n* = 16): ♂10; ♀6 69 (53–79) years 70.5 (47.5–95.9) kg 23.8 (18.1–30.8) kg/m² WL total: −7.9% (−33.0–3.1) WL last month: 3.95% (−12.9–3.1)	**IG**: oral EN as IN with target of 1.5 g protein/kg (per pack 16.8 g protein + 250 mL water) between meals (target). Consumption amounts should be recorded in diaries. Recording of other protein intake one week before via Questionnaire on consumption frequency. Estimated dosage (1–4 packs/d) (250–1000 mL: 16.8–67.2 g protein/d) (median intake at 2 pck./d). **CG**: standard of clinical care (screening with NRS-2002; individual counselling by nursing staff on the topic of food supplements and visit of a nutritionist before diagnosis).	7 days preOP	Overall postoperative complications and LOS
Gavazzi et al. (2016) [56]	OeCa, GaCa, PaCa, bile duct cancer *n* = 79 thereof PaCa *n* = 13	**IG**: (*n* = 38): ♂23; ♀15 67 (62–74) years **CG**: (*n* = 41): ♂26; ♀15 69 (58–76) years	**IG**: HEN according to energy requirements by Harris and Benedict. Overnight supplementation to oral diet with any polymeric standard diet containing 1–1.5 kcal/mL, 50–60% carb, 25–35% lipids, 12–20% proteins (target). HEN could be discontinued 2 months after surgery if 5% weight gain was recorded. **CG**: nutritional counselling by a dietician incl. total energy and protein requirements. If necessary, prescription of oral food supplements.	During oncologic treatment; 1 day postOP– regular oral intake.	Nutritional status (body weight, weight change)
Giger et al. (2007) [57]	GaCa, PaCa, periampullary cancer *n* = 46 thereof PaCa *n* = 30	**IG****1** (*n* = 14): ♂8; ♀6 64.4 (30–84) years 23.7 (±3.5) kg/m² **IG** **2** (*n* = 17): ♂10; ♀7 57.1 (33–77) years 23.3 (±4.0) kg/m² **CG** (*n* = 15): ♂9; ♀6 63.0 (47–79) years 22.7 (±3.3) kg/m²	**IG****1**: enteral IN (Impact^®^ (Novartis Consumer Health, Switzerland)) 5 days (1 L/d). **IG** **2**: enteral IN + dietary supplement glycine (Impact plus glycine^®^ (Novartis Consumer Health, Switzerland)) 2 days. **CG**: no preoperative nutritional support. **IG** **1** **and** **IG** **2**: received enteral IN for 7 days postOP, diet should provide 25 kcal/kg/d (target).	5–2 days preOP IN 1–7 days postOP suppl. or EN + suppl.	Postoperative serum level of C-reactive protein
Hamza et al. (2015) [58]	Periampullary cancer *n* = 37	**IG**: (*n* = 17): ♂9; ♀8 63 (58–69) years WL: 9.2 (6.8–11.6)% **CG**: (*n* = 20): ♂11; ♀9 67 (63–70) years WL: 9.6 (7.5–11.8)%	**IG**: EN: immune-boosting nutrition: Impact^®^ (oral) (Novartis Medical Nutrition, UK) with arginine, omega-3 fatty acids, mRNA. Both: provide 150 kcal/100 mL, non-isonitrogenous. Protein content Impact: 8.4 g/100 mL vs. 6.0 g/100 mL in standard diet due to addition of arginine and mRNA in Impact diet. **CG**: preOP EN with Fresubin^®^ (Fresenius Kabi Ltd., UK).	14 days preOP; 24 h to min. 7 days postOP	Parameters of systematic immune function (IL-1-α, TNF-α, total lymphocyte count (TLC), CD4, CD8, CD25, CD56, CH50, C3, C4)
Hyltander et al. (2005) [59]	OeCa, GaCa, PaCa, others *n* = 80 thereof PaCa *n* not reported	**IG****1** (*n* = 26): ♂18; ♀8 62 (±2) years 23.6 (±0.6) kg/m² WL: 5 (±0.4)% **IG** **2** (*n* = 27): ♂17; ♀10 62 (±2) years 23.8 (±0.6) kg/m² WL: 5 (±0.3)% **CG** (*n* = 27): ♂19; ♀9 63 (±2) years 23.8 (±0.5) kg/m² WL: 5 (±0.8)%	**IG****1**: EN + oral nutrition. **IG** 2: PN + oral nutrition. **CG**: oral nutrition. Composition of EN + PN(%kcal): 35 (±3)% fat; 39 (±4)% carb; 16 (±2)% protein; protein 0.9–1.1 g/kg BW/d. Pre- and postOP: counselling by dietician with implementation advice on energy intake, frequency of meals, liquid and solid foods. Oral nutritional supplements were offered (high energy and high protein were recommended). **IG** **1**: Impact^®^ (Novartis Nutrition) (1–10 days postOP). From 11 days postOP standard enteral formula with 1000 mL/d (Nutridrip standard^®^ (Novartis Nutrition)). **IG** **2**: Vitrimix^®^ (Fresenius Kabi) (900 kcal) incl. vitamins, minerals and trace elements **CG**: standard electrolyte solution. All patients in all groups received recommendations from nutritionists.	1–10 days postOP–EN; then EN with standard formula. Pre- and post-discharge: nutritional counselling	Recovery of nutritional state (body fat, lean body mass)
Jo et al. (2006) [60]	Periampullary cancer *n* = 60	**IG** (*n*= 32): ♂19; ♀13 56.8 (±9.4) years WL: 3.2 (0–20.5)% **CG** (*n* = 28): ♂10; ♀18 56.9 (±10.3) years WL: 5.9 (0–13.4)%	**IG**: PN as amino acid suppl. + glutamine (2.0 g/100 mL; 15% amino acid solution; target: 10 mL = 0.2 g glutamine/kg BW/d). **CG**: PN isonitrogenous (target: 1.3 g/kg BW amino acids/d); postOP supplemental (30 kcal/kg/d with 1.3 g/kg amino acids).	Day 2 preOP–5 days postOP	Patient’s discharge from hospital
Klek et al. (2008) [61]	GaCa, PaCa, *n* = 183 thereof PaCa *n* = 69	**IG**: (*n* = 92): ♂34; ♀14 62.3 (±11.3) years BMI <19 kg/m²: *n* = 17 BMI >19 kg/m²: *n* = 74 WL <10% (last 3–6 month): *n* = 73 WL >10% (last 3–6 month): *n* = 16 **CG**: (*n* = 91): ♂35; ♀13 62.1 (±10.9) years BMI <19 kg/m²: *n* = 10 BMI >19 kg/m²: *n* = 38 WL <10% (last 3–6 month): *n* = 76 WL >10% (last 3–6 month): *n* = 18	**IG**: enteral IN: Reconvan^®^ (Fresenius Kabi, Poland). **CG**: EN standard oligopeptide diet Peptisorb^®^ (Nutricia Ltd., Poland). Energy same; protein target (**CG**: 4.0 g vs. **IG** 5.5 g); total fat target (**CG**: 1.7 g vs. **IG**: 4.1 g) (SAFA: 1.0 vs. 3.3; of which MCT: 0.8 vs. 1.9); Sodium: 100 mg vs. 138; Potassium: 150 mg vs. 207. **CG**: standard oligopeptic EN.	6 h postOP–7 days postOP	Postoperative infectious complications
Klek et al. (2011) [62]	PaCa, GaCa, *n* = 305 thereof PaCa *n* = 94	**IG**: (*n* = 152): ♂92; ♀60 60.2 (±12.4) years 17.9 (±2.8) kg/m² WL: 18.3 (±4.4)% **CG**: (*n* = 153): ♂89; ♀62 [sic] 61.5 (±11.8) years 17.9 (±2.8) kg/m² WL: 18.8 (±4.9)%	6 h after surgery with 5% glucose solution for the first 12 h, following. **IG**: postOP enteral IN: Reconvan^®^” (Fresenius Kabi, Poland). **CG**: standard oligopeptide diet: infusion of Peptisorb^®^ (Nutricia Ltd. Poland).	6 h postOP until 7 days postOP	Number of complications
Krueger et al. 2016 [63]	Biliopancreatic lesions partly as PaCa *n* = 100	**IG**: (*n* = 51): ♂28; ♀23 69.5 (58.2–75.8) years 80.6 (69.8–87.8) kg 26.6 (24.3–32.0) kg/m² WL b. d.: 3.0 (12.0–0.0) kg **CG**: (*n* = 49): ♂29; ♀20 61.5 (55.6–71.3) years 75.6 (65.0–85.0) kg 25.3 (22.4–27.8) kg/m² WL b. d.: 5.0 (8.2–0.0) kg	**IG**: 1000 mL PN (target: 700 kcal, 25.3 g protein, 30 g fat, 75 g glucose + adapted nutrition with vitamins, trace elements on fasting days). **CG**: 1000 mL isotonic electrolyte solution on fasting days. Same daily oral energy intake during non-fasting days in hospital in **CG** and **IG** (1049 vs. 1082 kcal). Median suppl. PN (**IG**): 1400 kcal.	Undergoing in-hospital work-up for biliopancreatic mass lesions on fasting days	Body weight/ weight gain
Lobo et al. (2006) [64]	OeCa, GaCa, PaCa, *n* = 108 thereof PaCa *n* = 15	**IG**: (*n* = 54): ♂40; ♀14 65.7 (±1.4) years BMI <19 kg/m²: *n* = 4 BMI >19 kg/m²: *n* = 50 **CG**: (*n* = 54): ♂43; ♀11 66.6 (±1.4) years BMI <19 kg/m²: *n* = 5 BMI >19 kg/m²: *n* = 49	**IG**: experimental enteral IN. **CG**: isonitrogenous, isocaloric standard EN.	4 h postOP–10–15 days postOP	Development of infectious complications
Mori et al. (2019) [65]	PaCa *n* = 39	**IG**: (*n* = 19): ♂11; ♀8 66 (41–83) years 20.4 (15.0–26.2) kg/m² **CG**: (*n* = 20): ♂12; ♀8 64 (41–83) years 20.2 (17.7–29.9) kg/m²	Fat-free elemental EN (via jejunostomy tube). **IG**: EN until 3 months postOP (91 days (87–93)). **CG**: until adequate oral intake was achieved (10 days (8–45)). Composition: EN with target 600 kcal/d (2 doses in total) Elental^®^ (EA Pharma Co., Ltd., Japan) with 4.4 g protein/ 100 Kcal.	1 day postOP–3 months postOP	Complications necessitating readmission (postOP)
Moses et al. (2004) [66]	PaCa *n* = 24	**IG**: (*n* = 9): ♂6; ♀3 65 (±2) years 21 (±1) kg/m² WL: 21 (±2)% **CG**: (*n* = 15): ♂4; ♀11 70 (±3) years 20 (±1) kg/m² WL: 19 (±2)%	**IG****and****CG**: dietary supplement 8 weeks: 2 doses of 237 mL each containing 16 g protein, 311 kcal (target), both oral supplements ready-to-use, high calorie, high protein, low fat formulas which were isocaloric and isonitrogenic. **IG**: with omega-3 fatty acids (1.1 g EPA). **CG**: without omega-3 fatty acids; omega-3 fatty acids: balanced by omega-9 fatty acids.	Home-living PaCa patients (unresectable); 8 weeks	Body weight/ body composition
Park et al. (2012) [67]	PaCa, periampullary cancer *n* = 38 thereof PaCa *n* = 14	**IG** (*n* = 18): ♂7; ♀11 62.7 (±10.3) years 63.6 (±9.2) kg 23.8 (±3.9) kg/m² WL: 3.1 (±3.6) kg **CG** (*n* = 20): ♂12; ♀8 61.3 (±13.2) years 62.7 (±8.5) kg 23.5 (±2.1) kg/m² WL: 1.9 (±1.4) kg	**IG**: EN: target of 25 kcal/kg (18 h/d). **CG**: total PN: target of 25 kcal/kg/d. Solution ratio: glucose to lipids 2:1; non-protein to nitrogen (kcal/kg): 100:1. Total PN with vitamins, electrolytes, trace elements, insulin. 4 and 5 days postOP a sip of water. Within 7 days intake of a regular diet.	**IG**: 24 h postOP-oral intake >800 kcal/d **CG**: 1 day postOP-oral intake >800 kcal/ d	Change in weight
Perinel et al. (2016) [68]	PaCa *n* = 204	**IG****(PN)** (*n* = 101): ♂40; ♀61 64.02 (±9.90) years 23.76 (±3.44) kg/m² WL: 7.31 (±6.81)% **CG** **(EN)** (*n* = 103): ♂39; ♀64 65.46 (±11.25) years 24.99 (±4.17) kg/m² WL: 6.15 (±6.81)%	PreOP IN for all malnourished patients. **IG**: PN. **CG**: EN: isonitrogenous, isocaloric feeding (nasojejunale tube) at 25 mL/h from 1 day postOP. Amount increased by 25 mL/h every 24 h (administration over 20 h). **IG** **and** **CG** target: 30 kcal/kg/d with 1.5 g amino acids/kg/d. Carb-/amino acid-ratio: 3/2.	1 day postOP-oral food intake 60% of nutrient requirement	Incidence of postoperative complications
Slotwinski et al. (2008) [69]	PaCa *n* = 41	**IG**: (*n* = 19): ♂14; ♀5 59.8 (±6.0) years BMI preOP: 23.4 (±4.5) kg/m² BMI postOP: 22.4 (±6.3) kg/m² WL preOP: 6.5 (±2.1)% WL postOP: 9.1 (±2.8)% **CG**: (*n* = 22): ♂15; ♀7 54.2 (±4.1) years BMI preOP: 22.2 (±3.2) kg/m² BMI postOP: 21.8 (±3.0) kg/m² WL preOP: 6.3 (±3.4)% WL postOP: 9.2 (±3.2)%	**IG**: enteral IN: target: 14.7(±2.2)g nitrogen, 177(±26 g) glucose, 51.4(±7.5)g fat, 16.4(±2.4)g glutamine, 10.9(±1.6)g arginine (incl. 91.8(±13.5)g protein and 1529(±224) kcal). **CG**: standard EN: target: 10.8(±1.3) g nitrogen, 208(±24) g glucose, 66.0(±7.7) g fat (incl. 102(±12) g protein and 1693(±198) kcal). **IG****+****CG**: antibiotic as well as low-particle heparin, crystalline-line fluids intravenous and electrolytes as needed.	1–12.3 (±2.0) day postOP	Cellular immunity
Werner at al. (2017) [70]	PaCa *n* = 33	**IG**: (*n* = 18): ♂7; ♀11 70.3 (±8.24) years 21.3 (±1.73) kg/m² **CG**: (*n* = 15): ♂9; ♀6 71.3 (±7.51) years 23.7 (±4.10) kg/m²	**IG**: dietary supplement IN: FO capsule: 60% FO, 40% MCT (6.9 g/100 g EPA and 13.6 g/100 g DHA); target: 3 × 1 capsule. **CG**: dietary supplement IN: MPL capsule: 35% omega-3 fatty acids phospholipids (mainly phospatidylcholine) + 65% neutral lipids (8.5 g/100 g EPA and 12.3 g/100 g DHA); target: 3 × 1 capsule MPL and FO each 300 mg EPA and DHA/d.	During chemo-/radio-/supportive or alternative therapy (palliative and curative); 6 weeks	Change in weight and appetite

BMI: body mass index; BW: body weight; **CG**: control group; CoCa: colon cancer; DHA: docosahexaenoic acid; EN: enteral nutrition; EPA: eicosapentaenoic acid; FO: fish oil; GaCa: gastric cancer; HEN: home-enteral nutrition; **IG**: intervention group; IN: immunonutrition; LOS: length of stay in hospital; MCT: medium-chain triglyceride; MPL: marine phospholipids; n.a.: data not available; NI: nutritional intervention; OeCa: oesophagus cancer; PaCa: pancreatic cancer; PN: parenteral nutrition; RNA: ribonucleic acid; SAFA: saturated fatty acid; WL: weight loss.

**Table 2 cancers-14-02212-t002:** Randomized controlled trials of parenteral nutrition.

Reference	No. of Patients	Type of Cancer	Evaluated Outcome(s)	Summary of Results
**Nutritional status (WL, BIA parameters)**				
Krueger et al., 2016 [63]	100	Biliopancreatic lesions partly as PaCa	Body weight/ weight gain	No sign. difference in median weight change (IG: −0.2 (−1.4; 0.5) kg, CG: −0.6 (−1.7; 0.1) kg, *p* = 0.217 Sign. difference in weight change adjusted for nine potential influencing factors in a multivariate regression analysis (IG: +1.27 kg compared to CG, *p* = 0.027)
**Complications (infections, mortality, LOS)**				
Brennan et al., 1994 [51]	117	PaCa; periampullary cancer	Generic role of TPN	No sign. difference in median survival (IG and CG: 24 months, *p* = 0.25) No sign. difference in minor complications (IG: 32, CG: 24, *p* = 0.30) Sign. difference in major complications (IG: 27, CG: 13, *p* = 0.02) Sign. difference in severe complications (e.g., abscess) (IG: *n* = 12 vs. CG: *n*= 2, *p* = 0.01) No sign. difference in postOP mortality (CG: 2%; IG: 7%, *p* = 0.17) No sign. difference in reoperation (IG: *n* = 6, CG: *n*= 3, *p* = 0.18)
Jo et al. (2006) [60]	60	Periampullary cancer	Patients’ discharge from hospital	Trial discontinued: glutamine solution was no longer available. No sign. difference in LOS (IG: 14.0 (9–54) d; CG: 14.5 (9–41) d; *p* = 0.197)

BIA: bio impedance analysis; CG: control group; **IG**: intervention group; LOS: length of stay in hospital; PaCa: pancreatic cancer; TPN: total parenteral nutrition; WL: weight loss.

**Table 3 cancers-14-02212-t003:** Randomized controlled trials of enteral nutrition.

Reference	No. of Patients	Type of Cancer	Evaluated Outcome(s)	Summary of Results
**Nutritional status (WL, BIA parameters)**				
Gavazzi et al. (2016) [56]	79	OeCa, GaCa, PaCa, Bile duct cancer	Nutritional status	Study was discontinued due to advantage for IG. Sign. difference in change in body weight from baseline to 2 months: IG: −0.3 ± 3.9 kg (−0.5%); CG: −3.6 ± 4.8 kg (−5.8%); treatment effect 3.26 kg (*p* = 0.0031)
**Immunology (Cytokines, CRP, Albumin)**				
Slotwinski et al. (2008) [69]	41	PaCa	Cytokines	Sign. higher concentrations of IL-6 (day 10, *p* =0.017), IL-8 (day 1: *p* = 0.01; day 3, 7, 10: *p* < 0.001), IL-10 (day 3, 10: *p* < 0.001), IL-1RA (day 7: *p* < 0.001; day 10: *p* = 0.002) in IG Sign. higher postOP levels of IL-1ß (day 7: *p* < 0.001), TNF-α (day 3: *p* = 0.006; day 7: *p* < 0.001) in CG Sign. increase in total lymphocyte count (nutritional status) in IG (IG 1140 ± 262; CG: 930 ± 145 cells/mm³; *p* = 0.003)
Hamza et al. (2015) [58]	37	Periampullary cancer	Parameters of systematic immune function (IL-1-α, TNF-α)	No sign. difference in IL-1-α (IG: 69 (23–115) pg/mL vs. CG: 73 (29–117) pg/mL; *p* = 0.529) Sign. difference in TNF- α in IG over period of 14 days (day 14: 1993 pg/mL; day 0: 738 pg/mL; *p* = 0.047) No sign. difference in TNF-α (IG: 738 (482–993) pg/mL vs. CG: 1212 (647–1779) pg/mL; *p* = 0.112)
**Complications (Infections, mortality, LOS)**				
Braga et al. (1999) [49]	171	CoCa, GaCa, PaCa	Rate of postoperative infectious complications	Sign. difference in rate of complications (IG: 11%, CG: 24%; *p* = 0.02)
			LOS	Sign. difference in LOS (IG: 11.1 (± 4.4) days, CG: 12.9 (± 4.6) days; *p* = 0.01)
Daly et al. (1995) [52]	60	OeCa, GaCa, PaCa, others	Clinical outcome	Sign. difference in development of major postOP infections and wounds (IG 1 and 2: 3/30 patients (10%), CG 1 and 2: 13 / 30 patients (43%); *p* < 0.005) Sign. difference in LOS (IG 1 and 2: 16 (± 0.9) d, CG 1 and 2: 22 (±2.9) d; *p* < 0.005)
Klek et al. (2008) [61]	183	GaCa, PaCa	Postoperative complications	No sign. differences (IG: *n* = 21 (23.1%), CG: *n* = 23 (25.2%); *p* > 0.05)
Lobo et al. (2006) [64]	108	OeCa, GaCa, PaCa	Development of infectious complications	No sign. differences (IG: *n* = 24 (44.4%), CG: *n* = 24 (44.4%); *p* = 1.0)
Klek et al. (2011) [62]	303	PaCa, GaCa	Number of complications	Sign. differences in infectious complications (IG *n* = 43 (28.3%), CG: *n* = 60 (39.2%); *p* = 0.04)
Gade et al. (2016) [55]	35	PaCa	Overall postoperative complications	No sign. difference in postOP complications (data not shown; *p* > 0.05) Sign. larger number of participants with >3 postOP complications in CG (*p* = 0.030)
			LOS	No sign. difference in LOS (IG: 11 (6–30) days, CG: 16 (8–30) days; *p* = 0.549)
Mori et al. (2019) [65]	39	PaCa	Complications necessitating readmission postOP	Sign. difference in readmission rate at: o 90 days (IG: 0%, CG: 25%; *p* = 0.047) o 6 months postOP (IG: 5.3%, CG: 35.0%; *p* = 0.044) o 12 months postOP (IG: 10.5%, CG: 40.0%; *p* = 0.065) Sign. difference in cumulative readmission rate postOP (*p* = 0.018): o 3 months (IG: 0%, CG: 24.7%) o 6 months (IG: 5.3%, CG: 31.7%) o 12 months (IG: 12.6%, CG: 43.7%) o 24 months (IG: 25.1%, CG: 43.7%)

BIA: bio impedance analysis; CG: control group; CoCa: colon cancer; CRP: C-reactive protein; GaCa: gastric cancer; IG: intervention group; LOS: length of stay in hospital; OeCa: oesophagus cancer; PaCa: pancreatic cancer; POD: postoperative day; WL: weight loss.

**Table 4 cancers-14-02212-t004:** Randomized controlled trials of dietary supplements.

Reference	No. of Patients	Type of Cancer	Evaluated Outcome(s)	Summary of Results
**Nutritional status (WL, BIA parameters)**				
Douglass et al. (1978) [53]	30	PaCa, GaCa, CoCa	Weight loss/weight changes	Weight loss lower in IG with PaCa (IG: 3.5%, CG 6.4%; *p* not reported) Weight change higher in IG (IG: −6% (−2 to −12), CG: −5% (9 to −10))
Bauer et al. (2005) [47]	185	PaCa	Body composition	No sign. difference in change in body weight during 8 week increase in IG and decrease in CG (IG: +0.5 kg, CG: −0.7 kg; *p* = 0.052) Sign. difference in body weight (average after 8 weeks) (IG: 61.8 kg, CG: 60.0 kg; *p* < 0.0001) No sign. difference in lean body mass (IG: 44.1 kg, CG: 43.6 kg, *p* = 0.556)
Fearon et al. (2003) [54]	200	PaCa	Body weight and body composition	No sign. differences in weight loss (IG: −0.25/month, CG: −0.37 kg/month; *p* = 0.74) No sign. change in lean body mass (IG: 0.27 kg/month, CG: 0.12 kg/month; *p* = 0.88) Sign. correlation between supplement intake within IG and o weight gain (*p* < 0.001) o increase in lean body mass (*p* = 0.036)
Moses et al. (2004) [66]	24	PaCa	Body weight and body composition	No sign. difference in body weight after 8 w (IG: 0.0 (±1.3) kg, CG: −0.2 (±0.8) kg; *p* reported as NS) No sign. differences in lean body mass after 8 weeks (IG: 0.3 (±0.5) kg, CG: 0.6 (±0.8) kg, *p* reported as NS)
Werner at al. (2017) [70]	33	PaCa	Change in weight	Sign. weight stabilization after 6 weeks in comparison to the weight loss before the study within both groups (IG: *p* = 0.001, CG: *p* = 0.003)
**Immunology (Cytokine, CRP, Albumin)**				
Ashida et al. (2019) [46]	20	Periampullary cancer	Serum concentration of IL-6	no sign. differences in IL-6 after intervention (7 days preOP) (*p* = 0.68)
Braga et al. (2012) [50]	36	PaCa, periampullary cancer	Inflammatory response	no sign. differences in C-reactive protein levels (mg/L) after surgery between groups: o baseline: IG: 17.7 (±41.2), CG: 13.2 (±20.4); (*p* = 0.52) o POD 1: IG: 118.3 (±52.8), CG: 93.9 (±39.4); (*p* = 0.13) o POD 3: IG: 158.9 (±67.5), CG: 140.0 (±52.8); (*p* = 0.57) o POD 7: IG: 89.7 (±54.7), CG: 89.7 (±54.7); (*p* = 0.97)

BIA: bio impedance analysis; CG: control group; CoCa: colon cancer; CRP: C-reactive protein; GaCa: gastric cancer; IG: intervention group; NS: not significant; PaCa: pancreatic cancer; POD: postoperative day; WL: weight loss.

**Table 5 cancers-14-02212-t005:** Randomized controlled trials of mixed and special forms of nutritional interventions.

Reference	No. of Patients	Type of Cancer	Evaluated Outcome(s)	Summary of Results
**Nutritional status (WL, BIA parameters)**				
Park et al. (2012) [67]	38	PaCa, periampullary cancer	Change in weight	Sign. difference in BMI on postOP day 21 (IG: 23.7 (±5.1) kg/m²; CG: 21.8 (±2.1) kg/m²; *p* = 0.005)
Akita et al. (2019) [45]	62	PaCa	Ratio of skeletal muscle mass and psoas major muscle mass	No sign. difference in ratio of skeletal muscle mass and psoas major muscle area (IG:0.99 (±0.060), CG: 0.96 (±0.079); *p* = 0.102)
Hyltander et al. (2005) [59]	80	OeCa, GaCa, PaCa, others	Recovery of nutritional state	No sign. difference in body weight after 12 months (IG1: 64 ± 3 kg, IG2: 66 ± 4 kg, CG: 65 ±3 kg; *p* reported as NS) No sign. difference in body fat after 12 months (IG1: 13.5 ±1.5 kg, IG2: 15.7 ± 1.9 kg, CG: 13.4 ± 1.4 kg; *p* reported as NS) No sign. difference in lean body mass after 12 months (IG1: 49.7 ± 2.4 kg, IG2: 48.7 ± 2.4 kg, CG: 48.0 ± 2.1 kg; *p* reported as NS)
**Immunology (Cytokines, CRP, Albumin)**				
Giger et al. (2007) [57]	46	GaCa, PaCa, periampullary cancer	Postoperative serum level of CRP	Sign. difference in CRP on POD7 (IG1: 37.2 (±21.2) mg/L; IG2: 38.5 (±26.5) mg/L; CG: 93.0 (±17.3) mg/L (*p* < 0.05)) Sign. increase in all groups vs. baseline (*p* < 0.05)
**Complications (Infections, Mortality, LOS)**				
Bourdel-Marchasson et al. (2014) [48]	336	Colon (22.4%), lymphoma (14.9%), lung cancer (10.4%), abdominal PaCa (17.0%)	1-year mortality	No sign. difference in 1-year mortality (IG: 43.8%, CG: 41.3%; *p* = 0.74)
Perinel et al. (2016) [68]	204	PaCa	Incidence of postoperative complications	Sign. difference in the incidence of postOP complications (IG: 64.4%, CG: 77.5%; *p* = 0.040)
			Incidence of infectious complications	No sign. difference in the incidence of infectious complications (IG: 39.2%, CG: 41.6%; *p* = 0.731)

BIA: bio impedance analysis; CG: control group; CRP: C-reactive protein; GaCa: gastric cancer; IG: intervention group; LOS: length of stay in hospital; NS: not significant; OeCa: oesophagus cancer; PaCa: pancreatic cancer; POD: postoperative day; WL: weight loss.

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
