# Peer review of "Nutritional Interventions in Pancreatic Cancer: A Systematic Review"

_cancers, 2022, doi:10.3390/cancers14092212_

Round 1

Reviewer 1 Report

The authors perform a systematic review (NOT a meta-analysis) of reported clinical trials that looked at the benefits of nutritional interventions for pancreatic cancer patients, using weight gain/loss as a primary endpoint.

The authors report a significant heterogeneity in study designs, interventions, readouts, patient stratification and even in general criteria to define common endpoints (such as "cachexia"). This, along with possible technical bias detected in several studies, does not allow a firm conclusion and clinical recommendation but still allows the authors to outline interesting benefits of omega 3 supplements and enteral nutrition, compared to parenteral nutrition that is associated with frequent complications.

Author Response

Dear Reviewer 1

Thank you very much for the positive reviews of our manuscript Cancers-1680438: “Nutritional Interventions in Pancreatic Cancer: A Systematic Review”.

We are very pleased with your favorable rating and your recommendation for publication.

We have considered the comments of reviewer 3 and hope that the manuscript is now suitable for publication in Cancers.

Yours sincerely,

Ingeborg Rötzer

Comments and our answers reviewer 3:

Reviewer’s Comment

Answer

Simple Summary and Abstract: the conclusions are contradicting. In the summary, the authors suggested no concrete recommendations could be made; in the abstract, two specific nutrition intervention are recommended. Please make them consistent.

Thank you for your comment. The sentence in simple summary is replaced by:

Considering the heterogeneous study situation as well as the high bias potential of the included RCTs, a recommendation for enteral nutrition and dietary supplements with omega-3 fatty acids can be given.

Page 1 line 35: It is not clear what is 'lower complications and cytokine rates'.

We extended the sentence: lower rate and development of complications, positive effects on cytokine rates…

Page 1 line 36: What is 'better' body weight? For cancer patients the direction is not clear.

We replaced the word ,,better” by ,,higher “

Page 2 line 54-68:

The logic of this paragraph doesn't flow well. It begins with the definition of Cachexia, then mentioned the difference between Cachexia and Malnutrition is the inflammatory process. The definition of malnutrition is then described, which is a bit abrupt –I suggest moving this around to just after the definition of Cachexia, and then point to the differences. The authors should then mention the differences between the two, and discuss what are the specific inflammatory processes. It is also unclear if this review intends to cover both Cachexia and Malnutrition, or either one of them. Please state clearly.

Thank you for your recommendation. We have changed the order as you suggested. Further, we extended a sentence about the inflammatory process: ,,The systemic inflammation is triggered by cytokines and tumour derived factors. Increased serum CRP, IL6 and growth differentiation factor-15 are associated with cachexia.”

At the end of the introduction, we state that we are focusing on cachexia, malnutrition and weight loss.

Page 2 line 80 and page 3 line 105: it is unclear what 'data situation' refers to.

We extended the two sentences: The current data situation about NI in cachectic patients is referred to as ,,wildfire.

Because of the poor data situation regarding NI in PaCa, parameters associated with the nutritional status like immunology and complications should be included.

Page 2 lines 85-90: This sentence reads a bit redundant. A slight revision would be appreciated.

We replaced the word ,,focused” by ,,examined”.

Page 4 line 170: Quite a few included studies were added from manual search. I'm curious as to why these relevant studies did not appear in the initial search. I would suggest that the authors provide some justification (e.g. database indexing issues?) to reassure that the search strategy works for its purpose.

Thank you very much for your comment. Yes, we have added many studies by hand search, which has enabled us to improve the review. We followed the Cochrane standard, which recommends a supplementary hand search. The reason why so many studies were not found via the original search strategy could be that older studies are not always correctly indexed.  Furthermore, the choice of title and/or key words was not always helpful in older studies.

Page 5 line 180, page 14 line 338: Please spell out BMI when it first appears in the text, then use the abbreviation throughout.

We have corrected this.

Table 2 contains mass information, which is good, but the alphabetically ordered presentation makes it less reader-friendly. It could be a more useful summary table if the authors reorganized the references and group them by e.g. the outcomes, so that the readers may locate the relevant studies when looking for a specific outcome.

Thank you for your advice. We assume that you mean Table 1 (characteristics of included studies). We have followed Cochrane's scientific standard, which clearly presents the included studies in alphabetical order at the beginning of the results section. We can understand your argument and have therefore also decided to sort the results tables of the nutrition interventions according to defined outcomes and not according to alphabet or risk of bias.

The use of 'Complications (infections, mortality, LOS)' in Table 2, Table 3, and the main text should be revised because Complications does not logically contain mortality or LOS.

Thank you very much for your comment. Due to the heterogeneity of the outcomes, we have tried to define upper categories. We have described this in the introduction. The term complications should therefore be understood as an extended term. In order to give the reader information about what we understand by the defined categories, we have deliberately included all outcomes that belong to the generic terms in brackets in the results tables.

Page 10 line 219: ‘a tendency of higher mortality' is not supported by the studies above. Also, because the conclusions here were based on very few studies, I would not use a definitive tone. Instead, say something like 'parenteral nutrition appears unlikely to lead to weight change' and 'seemed to result in a higher chance of postoperative complications'.

We replaced the sentence by:

In summary, parenteral nutrition appears unlikely to lead to weight change and seems to result in a higher chance of postoperative complications.

Page 10 line 227: 'in our defined category nutritional status' is grammarly confusing. Same problem for other paragraphs.

We have changed this to "our category". It is important to us to show the reader that these are categories defined by us and that different terms have been combined under them.

Page 13 line 299, 302: 'One further trial' -> Another trial

We have corrected this.

Page 17 lines 434-439: Was adherence the only (main) reason for the heterogeneity? Could it be that the interventions were heterogeneous to begin with?

We extended this sentence:

The overview of the risk of bias (Figure 2) shows that the studies have different weaknesses. Often there were weaknesses in the randomization process.

Page 18 line 460: multiple interventions are fine, as long as they have a control group too.

We replaced the sentence by:

Further trials should focus on simple or multiple nutritional intervention compared with a control group.

Page 18 lines 462-476: Many of the previous reviews were mentioned in the introduction. The authors intended to do it differently and address some issues not solved in the other reviews. Here in the discussion, please reflect on how the present study complements the others, summarized the literature more comprehensively, and what new insights were obtained (i.e. to confirm that this review fulfilled its aims).

Thank you very much for your comment. We extended the sentence:

We conclude in our review that nutritional counselling should be given as a complementary measure and EN or immunonutrition with dietary supplements should be prioritized.

Page 19 line 498: "The significance of the data is therefore generally difficult." The message here is not very clear.

We replaced the word ,,significance” with ,,impact”.

Page 19 line 503: This section summarizes the limitations of the research, but is less informative. More content should be added, including the heterogeneity of the literature, risk of bias, etc. Potential remedies to these limitations should also be discussed.

We extended the sentence: The heterogeneous literature and number of trials with high risk of bias were further limitations.

Page 19 line 505: I agree. As I said previously, this could be concerning if it's due to the search strategy. Please discuss further.

We extended 2 sentences:

Searching more databases and using mesh terms could have improved the search. The heterogenous literature and number of trials with high risk of bias were further limitations.

Page 19 line 510: The independent participation of two reviewers in the whole process is required for systematic review, which is not an advantage and should be removed. Besides, more should be added. See comment for lines 462-476.

The sentence was removed:

A further strength of our systematic review was the entire research process, which was conducted by two independent persons, which led to an increase of the hit rate.

Page 19 lines 525-528: I would not claim that these interventions should be recommended based on such vague and inconclusive evidence.

We extended the sentence:

However, due to the heterogeneity of the data, clear recommendations for these interventions are not possible.

The last paragraph of the Discussion section should make more suggestions for future research, addressing the research gaps in parenteral nutrition, enteral nutrition, dietary supplements, and mixed or special forms.

Thank you for your comment. We extended the paragraph:

In order to have a larger data base in the field of nutrition interventions and to be able to derive evidence-based recommendations, more RCTs on PN, EN and dietary supplements should be conducted. Further, RCTs on anti-inflammatory supplements and anti-inflammatory diet should be planned.

Rearrange: For the seventh paragraph of the Discussion, the content on nutritional counseling should be merged with the fifth paragraph, as they both belong to the discussion on mixed and special forms of nutritional interventions. The rest of the seventh paragraph discusses the causes of risk of bias, which should be included in the discussion of risk of bias in the eighth paragraph.

We have corrected this.

Reviewer 2 Report

Point 1. The paper is an exhaustive review on the role of nutrition in pancreatic cancer

Point 2. In my opinion it is relevant both scientifically and for the journal

Point 3. In addition to what is written in point 1, this review is updated to date while the others published in the medical literature are updated to 2016.

Point 4. the conclusions consistent with the evidence and arguments
presented and they address the main question posed

There are no specific comments. Authors presenta a very sound and interesting review of the role of nutrition in pancreatic adenocarcinoma. The Prisma methodology is correct and in my opinion the paper can be accepted in the present form. 

Author Response

Dear Reviewer 2

Thank you very much for the positive reviews of our manuscript Cancers-1680438: “Nutritional Interventions in Pancreatic Cancer: A Systematic Review”.

We are very pleased with your favorable rating and your recommendation for publication.

We have considered the comments of reviewer 3 and hope that the manuscript is now suitable for publication in Cancers.

Yours sincerely,

Ingeborg Rötzer

Comments and answers reviewer 3:

Reviewer’s Comment

Answer

Simple Summary and Abstract: the conclusions are contradicting. In the summary, the authors suggested no concrete recommendations could be made; in the abstract, two specific nutrition intervention are recommended. Please make them consistent.

Thank you for your comment. The sentence in simple summary is replaced by:

Considering the heterogeneous study situation as well as the high bias potential of the included RCTs, a recommendation for enteral nutrition and dietary supplements with omega-3 fatty acids can be given.

Page 1 line 35: It is not clear what is 'lower complications and cytokine rates'.

We extended the sentence: lower rate and development of complications, positive effects on cytokine rates…

Page 1 line 36: What is 'better' body weight? For cancer patients the direction is not clear.

We replaced the word ,,better” by ,,higher “

Page 2 line 54-68:

The logic of this paragraph doesn't flow well. It begins with the definition of Cachexia, then mentioned the difference between Cachexia and Malnutrition is the inflammatory process. The definition of malnutrition is then described, which is a bit abrupt –I suggest moving this around to just after the definition of Cachexia, and then point to the differences. The authors should then mention the differences between the two, and discuss what are the specific inflammatory processes. It is also unclear if this review intends to cover both Cachexia and Malnutrition, or either one of them. Please state clearly.

Thank you for your recommendation. We have changed the order as you suggested. Further, we extended a sentence about the inflammatory process: ,,The systemic inflammation is triggered by cytokines and tumour derived factors. Increased serum CRP, IL6 and growth differentiation factor-15 are associated with cachexia.”

At the end of the introduction, we state that we are focusing on cachexia, malnutrition and weight loss.

Page 2 line 80 and page 3 line 105: it is unclear what 'data situation' refers to.

We extended the two sentences: The current data situation about NI in cachectic patients is referred to as ,,wildfire.

Because of the poor data situation regarding NI in PaCa, parameters associated with the nutritional status like immunology and complications should be included.

Page 2 lines 85-90: This sentence reads a bit redundant. A slight revision would be appreciated.

We replaced the word ,,focused” by ,,examined”.

Page 4 line 170: Quite a few included studies were added from manual search. I'm curious as to why these relevant studies did not appear in the initial search. I would suggest that the authors provide some justification (e.g. database indexing issues?) to reassure that the search strategy works for its purpose.

Thank you very much for your comment. Yes, we have added many studies by hand search, which has enabled us to improve the review. We followed the Cochrane standard, which recommends a supplementary hand search. The reason why so many studies were not found via the original search strategy could be that older studies are not always correctly indexed.  Furthermore, the choice of title and/or key words was not always helpful in older studies.

Page 5 line 180, page 14 line 338: Please spell out BMI when it first appears in the text, then use the abbreviation throughout.

We have corrected this.

Table 2 contains mass information, which is good, but the alphabetically ordered presentation makes it less reader-friendly. It could be a more useful summary table if the authors reorganized the references and group them by e.g. the outcomes, so that the readers may locate the relevant studies when looking for a specific outcome.

Thank you for your advice. We assume that you mean Table 1 (characteristics of included studies). We have followed Cochrane's scientific standard, which clearly presents the included studies in alphabetical order at the beginning of the results section. We can understand your argument and have therefore also decided to sort the results tables of the nutrition interventions according to defined outcomes and not according to alphabet or risk of bias.

The use of 'Complications (infections, mortality, LOS)' in Table 2, Table 3, and the main text should be revised because Complications does not logically contain mortality or LOS.

Thank you very much for your comment. Due to the heterogeneity of the outcomes, we have tried to define upper categories. We have described this in the introduction. The term complications should therefore be understood as an extended term. In order to give the reader information about what we understand by the defined categories, we have deliberately included all outcomes that belong to the generic terms in brackets in the results tables.

Page 10 line 219: ‘a tendency of higher mortality' is not supported by the studies above. Also, because the conclusions here were based on very few studies, I would not use a definitive tone. Instead, say something like 'parenteral nutrition appears unlikely to lead to weight change' and 'seemed to result in a higher chance of postoperative complications'.

We replaced the sentence by:

In summary, parenteral nutrition appears unlikely to lead to weight change and seems to result in a higher chance of postoperative complications.

Page 10 line 227: 'in our defined category nutritional status' is grammarly confusing. Same problem for other paragraphs.

We have changed this to "our category". It is important to us to show the reader that these are categories defined by us and that different terms have been combined under them.

Page 13 line 299, 302: 'One further trial' -> Another trial

We have corrected this.

Page 17 lines 434-439: Was adherence the only (main) reason for the heterogeneity? Could it be that the interventions were heterogeneous to begin with?

We extended this sentence:

The overview of the risk of bias (Figure 2) shows that the studies have different weaknesses. Often there were weaknesses in the randomization process.

Page 18 line 460: multiple interventions are fine, as long as they have a control group too.

We replaced the sentence by:

Further trials should focus on simple or multiple nutritional intervention compared with a control group.

Page 18 lines 462-476: Many of the previous reviews were mentioned in the introduction. The authors intended to do it differently and address some issues not solved in the other reviews. Here in the discussion, please reflect on how the present study complements the others, summarized the literature more comprehensively, and what new insights were obtained (i.e. to confirm that this review fulfilled its aims).

Thank you very much for your comment. We extended the sentence:

We conclude in our review that nutritional counselling should be given as a complementary measure and EN or immunonutrition with dietary supplements should be prioritized.

Page 19 line 498: "The significance of the data is therefore generally difficult." The message here is not very clear.

We replaced the word ,,significance” with ,,impact”.

Page 19 line 503: This section summarizes the limitations of the research, but is less informative. More content should be added, including the heterogeneity of the literature, risk of bias, etc. Potential remedies to these limitations should also be discussed.

We extended the sentence: The heterogeneous literature and number of trials with high risk of bias were further limitations.

Page 19 line 505: I agree. As I said previously, this could be concerning if it's due to the search strategy. Please discuss further.

We extended 2 sentences:

Searching more databases and using mesh terms could have improved the search. The heterogenous literature and number of trials with high risk of bias were further limitations.

Page 19 line 510: The independent participation of two reviewers in the whole process is required for systematic review, which is not an advantage and should be removed. Besides, more should be added. See comment for lines 462-476.

The sentence was removed:

A further strength of our systematic review was the entire research process, which was conducted by two independent persons, which led to an increase of the hit rate.

Page 19 lines 525-528: I would not claim that these interventions should be recommended based on such vague and inconclusive evidence.

We extended the sentence:

However, due to the heterogeneity of the data, clear recommendations for these interventions are not possible.

The last paragraph of the Discussion section should make more suggestions for future research, addressing the research gaps in parenteral nutrition, enteral nutrition, dietary supplements, and mixed or special forms.

Thank you for your comment. We extended the paragraph:

In order to have a larger data base in the field of nutrition interventions and to be able to derive evidence-based recommendations, more RCTs on PN, EN and dietary supplements should be conducted. Further, RCTs on anti-inflammatory supplements and anti-inflammatory diet should be planned.

Rearrange: For the seventh paragraph of the Discussion, the content on nutritional counseling should be merged with the fifth paragraph, as they both belong to the discussion on mixed and special forms of nutritional interventions. The rest of the seventh paragraph discusses the causes of risk of bias, which should be included in the discussion of risk of bias in the eighth paragraph.

We have corrected this.

Reviewer 3 Report

In the manuscript titled ‘Nutritional Interventions in Pancreatic Cancer: A Systematic Review’, the authors reported on a systematic review of the literature on the effects of different nutritional interventions (parenteral nutrition, enteral nutrition, dietary supplements, mixed or special forms) in Pancreatic Cancer patients on weight loss, malnutrition and clinical indicators (Complications, infections, Mortality, LOS). The results could provide guidance for clinical practice, however, there are a few issues that have to be addressed.

  1. Simple Summary and Abstract: the conclusions are contradicting. In the summary, the authors suggested no concrete recommendations could be made; in the abstract, two specific nutrition intervention are recommended. Please make them consistent.
  2. Page 1 line 35: It is not clear what is 'lower complications and cytokine rates'.
  3. Page 1 line 36: What is 'better' body weight? For cancer patients the direction is not clear.
  4. Page 2 line 54-68: The logic of this paragraph doesn't flow well. It begins with the definition of Cachexia, then mentioned the difference between Cachexia and Malnutrition is the inflammatory process. The definition of malnutrition is then described, which is a bit abrupt - I suggest moving this around to just after the definition of Cachexia, and then point to the differences. The authors should then mention the differences between the two, and discuss what are the specific inflammatory processes. It is also unclear if this review intends to cover both Cachexia and Malnutrition, or either one of them. Please state clearly.
  5. Page 2 line 80 and page 3 line 105: it is unclear what 'data situation' refers to.
  6. Page 2 lines 85-90: This sentence reads a bit redundant. A slight revision would be appreciated.
  7. Page 4 line 170: Quite a few included studies were added from manual search. I'm curious as to why these relevant studies did not appear in the initial search. I would suggest that the authors provide some justification (e.g. database indexing issues?) to reassure that the search strategy works for its purpose.
  8. Page 5 line 180, page 14 line 338: Please spell out BMI when it first appears in the text, then use the abbreviation throughout.
  9. Table 2 contains mass information, which is good, but the alphabetically ordered presentation makes it less reader-friendly. It could be a more useful summary table if the authors reorganized the references and group them by e.g. the outcomes, so that the readers may locate the relevant studies when looking for a specific outcome.
  10. The use of 'Complications (infections, mortality, LOS)' in Table 2, Table 3, and the main text should be revised because Complications does not logically contain mortality or LOS.
  11. Page 10 line 219: ‘a tendency of higher mortality' is not supported by the studies above. Also, because the conclusions here were based on very few studies, I would not use a definitive tone. Instead, say something like 'parenteral nutrition appears unlikely to lead to weight change' and 'seemed to result in a higher chance of postoperative complications'.
  12. Page 10 line 227: 'in our defined category nutritional status' is grammarly confusing. Same problem for other paragraphs.
  13. Page 13 line 299, 302: 'One further trial' -> Another trial
  14. Page 17 lines 434-439: Was adherence the only (main) reason for the heterogeneity? Could it be that the interventions were heterogeneous to begin with?
  15. Page 18 line 460: multiple interventions are fine, as long as they have a control group too.
  16. Page 18 lines 462-476: Many of the previous reviews were mentioned in the introduction. The authors intended to do it differently and address some issues not solved in the other reviews. Here in the discussion, please reflect on how the present study complements the others, summarized the literature more comprehensively, and what new insights were obtained (i.e. to confirm that this review fulfilled its aims).
  17. Page 19 line 498: "The significance of the data is therefore generally difficult." The message here is not very clear.
  18. Page 19 line 503: This section summarizes the limitations of the research, but is less informative. More content should be added, including the heterogeneity of the literature, risk of bias, etc. Potential remedies to these limitations should also be discussed.
  19. Page 19 line 505: I agree. As I said previously, this could be concerning if it's due to the search strategy. Please discuss further.
  20. Page 19 line 510: The independent participation of two reviewers in the whole process is required for systematic review, which is not an advantage and should be removed. Besides, more should be added. See comment for lines 462-476.
  21. Page 19 lines 525-528: I would not claim that these interventions should be recommended based on such vague and inconclusive evidence.
  22. The last paragraph of the Discussion section should make more suggestions for future research, addressing the research gaps in parenteral nutrition, enteral nutrition, dietary supplements, and mixed or special forms.
  23. Rearrange: For the seventh paragraph of the Discussion, the content on nutritional counseling should be merged with the fifth paragraph, as they both belong to the discussion on mixed and special forms of nutritional interventions. The rest of the seventh paragraph discusses the causes of risk of bias, which should be included in the discussion of risk of bias in the eighth paragraph.

Author Response

Dear Reviewer 3

Thank you very much for the overall positive and helpful comments in our manuscript Cancers-1680438: “Nutritional Interventions in Pancreatic Cancer: A Systematic Review”.

We addressed all of your points and feel that this improved the manuscript. Point-by-point responses to your comments are attached and changes in the manuscript are highlighted with "track changes".

Yours sincerely,

Ingeborg Rötzer

Comments and answers:

Reviewer’s Comment

Answer

Simple Summary and Abstract: the conclusions are contradicting. In the summary, the authors suggested no concrete recommendations could be made; in the abstract, two specific nutrition intervention are recommended. Please make them consistent.

Thank you for your comment. The sentence in simple summary is replaced by:

Considering the heterogeneous study situation as well as the high bias potential of the included RCTs, a recommendation for enteral nutrition and dietary supplements with omega-3 fatty acids can be given.

Page 1 line 35: It is not clear what is 'lower complications and cytokine rates'.

We extended the sentence: lower rate and development of complications, positive effects on cytokine rates…

Page 1 line 36: What is 'better' body weight? For cancer patients the direction is not clear.

We replaced the word ,,better” by ,,higher “

Page 2 line 54-68:

The logic of this paragraph doesn't flow well. It begins with the definition of Cachexia, then mentioned the difference between Cachexia and Malnutrition is the inflammatory process. The definition of malnutrition is then described, which is a bit abrupt –I suggest moving this around to just after the definition of Cachexia, and then point to the differences. The authors should then mention the differences between the two, and discuss what are the specific inflammatory processes. It is also unclear if this review intends to cover both Cachexia and Malnutrition, or either one of them. Please state clearly.

Thank you for your recommendation. We have changed the order as you suggested. Further, we extended a sentence about the inflammatory process: ,,The systemic inflammation is triggered by cytokines and tumour derived factors. Increased serum CRP, IL6 and growth differentiation factor-15 are associated with cachexia.”

At the end of the introduction, we state that we are focusing on cachexia, malnutrition and weight loss.

Page 2 line 80 and page 3 line 105: it is unclear what 'data situation' refers to.

We extended the two sentences: The current data situation about NI in cachectic patients is referred to as ,,wildfire.

Because of the poor data situation regarding NI in PaCa, parameters associated with the nutritional status like immunology and complications should be included.

Page 2 lines 85-90: This sentence reads a bit redundant. A slight revision would be appreciated.

We replaced the word ,,focused” by ,,examined”.

Page 4 line 170: Quite a few included studies were added from manual search. I'm curious as to why these relevant studies did not appear in the initial search. I would suggest that the authors provide some justification (e.g. database indexing issues?) to reassure that the search strategy works for its purpose.

Thank you very much for your comment. Yes, we have added many studies by hand search, which has enabled us to improve the review. We followed the Cochrane standard, which recommends a supplementary hand search. The reason why so many studies were not found via the original search strategy could be that older studies are not always correctly indexed.  Furthermore, the choice of title and/or key words was not always helpful in older studies.

Page 5 line 180, page 14 line 338: Please spell out BMI when it first appears in the text, then use the abbreviation throughout.

We have corrected this.

Table 2 contains mass information, which is good, but the alphabetically ordered presentation makes it less reader-friendly. It could be a more useful summary table if the authors reorganized the references and group them by e.g. the outcomes, so that the readers may locate the relevant studies when looking for a specific outcome.

Thank you for your advice. We assume that you mean Table 1 (characteristics of included studies). We have followed Cochrane's scientific standard, which clearly presents the included studies in alphabetical order at the beginning of the results section. We can understand your argument and have therefore also decided to sort the results tables of the nutrition interventions according to defined outcomes and not according to alphabet or risk of bias.

The use of 'Complications (infections, mortality, LOS)' in Table 2, Table 3, and the main text should be revised because Complications does not logically contain mortality or LOS.

Thank you very much for your comment. Due to the heterogeneity of the outcomes, we have tried to define upper categories. We have described this in the introduction. The term complications should therefore be understood as an extended term. In order to give the reader information about what we understand by the defined categories, we have deliberately included all outcomes that belong to the generic terms in brackets in the results tables.

Page 10 line 219: ‘a tendency of higher mortality' is not supported by the studies above. Also, because the conclusions here were based on very few studies, I would not use a definitive tone. Instead, say something like 'parenteral nutrition appears unlikely to lead to weight change' and 'seemed to result in a higher chance of postoperative complications'.

We replaced the sentence by:

In summary, parenteral nutrition appears unlikely to lead to weight change and seems to result in a higher chance of postoperative complications.

Page 10 line 227: 'in our defined category nutritional status' is grammarly confusing. Same problem for other paragraphs.

We have changed this to "our category". It is important to us to show the reader that these are categories defined by us and that different terms have been combined under them.

Page 13 line 299, 302: 'One further trial' -> Another trial

We have corrected this.

Page 17 lines 434-439: Was adherence the only (main) reason for the heterogeneity? Could it be that the interventions were heterogeneous to begin with?

We extended this sentence:

The overview of the risk of bias (Figure 2) shows that the studies have different weaknesses. Often there were weaknesses in the randomization process.

Page 18 line 460: multiple interventions are fine, as long as they have a control group too.

We replaced the sentence by:

Further trials should focus on simple or multiple nutritional intervention compared with a control group.

Page 18 lines 462-476: Many of the previous reviews were mentioned in the introduction. The authors intended to do it differently and address some issues not solved in the other reviews. Here in the discussion, please reflect on how the present study complements the others, summarized the literature more comprehensively, and what new insights were obtained (i.e. to confirm that this review fulfilled its aims).

Thank you very much for your comment. We extended the sentence:

We conclude in our review that nutritional counselling should be given as a complementary measure and EN or immunonutrition with dietary supplements should be prioritized.

Page 19 line 498: "The significance of the data is therefore generally difficult." The message here is not very clear.

We replaced the word ,,significance” with ,,impact”.

Page 19 line 503: This section summarizes the limitations of the research, but is less informative. More content should be added, including the heterogeneity of the literature, risk of bias, etc. Potential remedies to these limitations should also be discussed.

We extended the sentence: The heterogeneous literature and number of trials with high risk of bias were further limitations.

Page 19 line 505: I agree. As I said previously, this could be concerning if it's due to the search strategy. Please discuss further.

We extended 2 sentences:

Searching more databases and using mesh terms could have improved the search. The heterogenous literature and number of trials with high risk of bias were further limitations.

Page 19 line 510: The independent participation of two reviewers in the whole process is required for systematic review, which is not an advantage and should be removed. Besides, more should be added. See comment for lines 462-476.

The sentence was removed:

A further strength of our systematic review was the entire research process, which was conducted by two independent persons, which led to an increase of the hit rate.

Page 19 lines 525-528: I would not claim that these interventions should be recommended based on such vague and inconclusive evidence.

We extended the sentence:

However, due to the heterogeneity of the data, clear recommendations for these interventions are not possible.

The last paragraph of the Discussion section should make more suggestions for future research, addressing the research gaps in parenteral nutrition, enteral nutrition, dietary supplements, and mixed or special forms.

Thank you for your comment. We extended the paragraph:

In order to have a larger data base in the field of nutrition interventions and to be able to derive evidence-based recommendations, more RCTs on PN, EN and dietary supplements should be conducted. Further, RCTs on anti-inflammatory supplements and anti-inflammatory diet should be planned.

Rearrange: For the seventh paragraph of the Discussion, the content on nutritional counseling should be merged with the fifth paragraph, as they both belong to the discussion on mixed and special forms of nutritional interventions. The rest of the seventh paragraph discusses the causes of risk of bias, which should be included in the discussion of risk of bias in the eighth paragraph.

We have corrected this.

Round 2

Reviewer 3 Report

Thank you for the thorough revision. Happy to know that my suggestions have been helpful!